# VitalSign^6^: A Primary Care First (PCP-First) Model for Universal Screening and Measurement-Based Care for Depression

**DOI:** 10.3390/ph12020071

**Published:** 2019-05-14

**Authors:** Madhukar H. Trivedi, Manish K. Jha, Farra Kahalnik, Ronny Pipes, Sara Levinson, Tiffany Lawson, A. John Rush, Joseph M. Trombello, Bruce Grannemann, Corey Tovian, Robert Kinney, E. Will Clark, Tracy L. Greer

**Affiliations:** 1Department of Psychiatry, University of Texas Southwestern Medical Center, Dallas, TX 75390, USA; manish.jha@utsouthwestern.edu (M.K.J.); farra.kahalnik@utsouthwestern.edu (F.K.); ronny.pipes@utsouthwestern.edu (R.P.); joseph.trombello@utsouthwestern.edu (J.M.T.); robert.kinney@utsouthwestern.edu (R.K.); edward.clark@utsouthwestern.edu (E.W.C.); tracy.greer@utsouthwestern.edu (T.L.G.); 2Icahn School of Medicine at Mount Sinai, New York City, NY 10029, USA; manish.jha@mssm.edu; 3Delivery System Improvement, University of Texas Southwestern Medical Center, Dallas, TX 75390, USA; sara.levinson@utsouthwestern.edu (S.L.); tiffany.lawson@utsouthwestern.edu (T.L.); 4Department of Psychiatry, Duke Medical School, Durham, NC 27710, USA; augustus.rush@duke.edu; 5Department of Psychiatry, Texas Tech Health Sciences Center, Permian Basin, TX 79763, USA; 6Duke-National University of Singapore, Singapore 168753, Singapore; 7Office of Communications, Marketing, and Public Affairs, University of Texas Southwestern Medical Center, Dallas, TX 75390, USA; corey.tovian@utsouthwestern.edu

**Keywords:** depression, screening, measurement-based care, primary care, mental health

## Abstract

Major depressive disorder affects one in five adults in the United States. While practice guidelines recommend universal screening for depression in primary care settings, clinical outcomes suffer in the absence of optimal models to manage those who screen positive for depression. The current practice of employing additional mental health professionals perpetuates the assumption that primary care providers (PCP) cannot effectively manage depression, which is not feasible, due to the added costs and shortage of mental health professionals. We have extended our previous work, which demonstrated similar treatment outcomes for depression in primary care and psychiatric settings, using measurement-based care (MBC) by developing a model, called Primary Care First (PCP-First), that empowers PCPs to effectively manage depression in their patients. This model incorporates health information technology tools, through an electronic health records (EHR) integrated web-application and facilitates the following five components: (1) Screening (2) diagnosis (3) treatment selection (4) treatment implementation and (5) treatment revision. We have implemented this model as part of a quality improvement project, called VitalSign^6^, and will measure its success using the Reach, Efficacy, Adoption, Implementation, and Maintenance (RE-AIM) framework. In this report, we provide the background and rationale of the PCP-First model and the operationalization of VitalSign^6^ project.

## 1. Introduction

This report describes the rationale, design, and implementation process of a quality improvement project called VitalSign^6^, which has focused on primary care as the initial access point of screening and treatment of depression. By recognizing, and effectively treating, clinical depression in primary care settings, the VitalSign^6^ project aims to improve access to, and quality of, behavioral healthcare, especially in minority and/or low income/uninsured patients.

## 2. What Is the Problem?

Major Depressive Disorder (MDD) is a chronic, often recurrent, and disabling [1,2] condition that affects 5–10% of adults in the United States every year [3,4,5]. In primary care, MDD is estimated to affect 10–14% of patients [6], yet rates of detection are low. By some estimates, over half of the MDD patients in medical settings go unrecognized [6,7]. For those cases that are diagnosed, treatment is often delayed, with a median time to treatment initiation of eight years [8]. For those that are treated with antidepressants, only one out of five receive adequate treatment (at least four visits with a physician in a 12-month period [4], or prescription of antidepressant medication exceeding the minimum effective dose for 2 or more months in a 12-month period [9]). The prescription of low doses of antidepressant medications [10], poor adherence to prescribed medications [11], and high drop-out rates early in treatment initiation [12] all contribute to poor outcomes in primary care settings. Estimates suggest only 6% of MDD patients achieve remission with acute-phase treatment [13].

Despite this apparent under-recognition and under-treatment, antidepressants and anxiolytics are among the most commonly prescribed medications in ambulatory care settings [14]. Additionally, studies of diagnoses rendered in electronic health records suggest moderate inter-rater reliability (median kappa below 0.5) with research diagnoses of depression [15]. Furthermore, about a third of prescriptions for antidepressants are inappropriate, due either to, off-label use, or prescription without strong scientific evidence, (no psychiatric diagnosis, lack of consideration for drug interactions and comorbidity) [16,17,18]. Better outcomes will require better recognition, more accurate diagnoses, and selection of appropriate medications, adequate dosing, and more persistent treatment that adheres to evidence-based guidelines.

## 3. How Is This Problem Being Addressed? 

Over the last two decades, several United States Preventive Services Task Force (USPSTF) and Agency for Healthcare Research and Quality (AHRQ) guidelines have recommended routine screening for depression among adults in health care settings to address the under-recognition of depression [19,20]. The Centers for Medicare & Medicaid Services (CMS) have also incentivized the incorporation of routine depression screening in primary care practices [21]. In response, some health systems have implemented large-scale formal screening programs [22]. However, screening alone is not sufficient; restricted access to mental health services due to a severe shortage of providers in the United States [23] has contributed to the lack of adequate follow-up for those patients who screen positive for depression. 

Poor access to behavioral health resources has led to the development of several models that provide depression screening along with follow-up care in primary care settings. Among these initiatives, one of the most studied is the Improving Mood-Promoting Access to Collaborative Treatment (IMPACT) collaborative care management program [24]. The collaborative care model involves coordination between a primary care provider, a behavioral health care manager, and a psychiatrist [25,26]. Other initiatives aimed to address depression in primary care have included (1) patients completing the 9-item Patient Health Questionnaire (PHQ-9) depression screening tool on paper and providing the physicians with these completed PHQ-9 questionnaires [27], (2) providers consulting with psychiatrists [10], (3) case management by health care assistants [28], and (4) screening followed by consultation (telephone and in-person) with pharmacists, based on published treatment guidelines [29]. The U.S. Department of Veteran’s Affairs (VA) has also launched large scale initiatives over the last decade to implement Primary Care—Mental Health Integration (PC-MHI) programs, which focus on co-location (primary care and psychiatric providers in the same physical location) as well as the collaborative care model [30,31].

## 4. How Have These Efforts Fared?

These efforts have resulted in limited success so far. Depression screening rates continue to be very low. According to the 2012 U.S. National Ambulatory Medical Care Survey (NAMCS), depression screening was conducted or ordered during only 1.4% (SE = 0.2%) of all primary care visits (a decline from 2.3% in 2010 NAMCS) [14]. Furthermore, neither consultation with pharmacists [29] nor consultation with psychiatrists resulted in improved outcomes as compared to treatment as usual [10]. Additionally, screening patients for depression and providing the screening results to physicians did not increase the active management of depression by physicians or decrease the depression severity during a six-month follow-up period [27].

On the other hand, case management by health care assistants, which included structured assessments of depression severity, support for adherence to medications, and timely feedback to treatment physicians, was associated with improved adherence and reduced depression severity over a 12-month follow-up period in a randomized controlled trial [28]. In the VA system, the PC-MHI program has been associated with higher rates of treatment initiation [32] and continuation [33], as compared to primary care services alone. In randomized controlled trials, the collaborative care (IMPACT) model has been shown to result in greater reductions in depressive symptoms, as compared to usual care, in primary care patients [24,34]. In the 2012 Cochrane review, based on seventy-nine randomized controlled trials, collaborative care was shown to be superior than usual care in the improvement of depression over the short, medium, and long-term [35]. However, efforts to implement collaborative care models in real-world clinical settings have not shown any significant advantage over usual care. The large-scale Depression Improvement Across Minnesota-Offering a New Direction (DIAMOND) study, which implemented a collaborative care model in 75 primary care clinics, found no significant difference in remission rates between clinics providing usual care versus those that utilized collaborative care [36]. To date, the evidence indicates that while the collaborative care model is successful in randomized controlled trials, when implemented in real-world settings, it shows no significant improvement in clinical outcomes. In addition, this model requires additional mental health staff, entails higher costs, and risks diffusion of responsibility between primary care providers and mental health professionals. Taken together, these results indicate that current models have not been effective in improving the process of care delivery for depressed patients in primary care settings. 

## 5. An Alternative Approach

We contend that a systematic approach to tailoring treatment to individuals, using the principles of measurement-based care (MBC), with treatment algorithms geared toward maximizing symptom reduction, minimizing side effects, monitoring adherence, and achieving recovery [37] will exceed the effects of treatment as usual [38] and produce substantially higher rates of remission [39]. This contention is supported by our prior work that has shown that treatment outcomes are comparable between primary and psychiatric care settings under pragmatic trial conditions [40,41]. Our Primary Care First (PCP-First) approach is centered on empowering the patients and providers to initiate a collaborative, patient-provider partnership, to improve depression treatment in primary care by incorporating the following activities: *Recognition* of depression through universal screening.*Diagnosis* of depression using DSM-5 diagnostic criteria.*Selection of appropriate treatment* (active surveillance, brief therapy, medication management) based on more accurate clinical evaluation.*Tailored medication delivery* using Measurement-Based Care (e.g., optimized, algorithm-based dosing with decision support/consultation, based on regular, systematic assessment of symptoms, side-effects, and adherence).*Continuation/maintenance phase* treatment.*System-level monitoring of adherence to evidence-based treatment recommendations/guidelines* via feedback to clinicians (i.e., clinical decision support, rounds with consulting clinicians, patient navigation) at the point of care.

Using this approach, the PCP takes primary responsibility for depression care, leveraging the already established provider-patient relationship, and the provider expertise in evidence-based care for chronic disorders. Additionally, this approach does not require major changes in roles and may be more cost-effective by reducing or eliminating the need to hire additional personnel for clinical care.

## 6. Incorporating Health IT Advances into Depression Care

We believe that advances in health information technology (health-IT) can provide elements critical for the success of the patient-provider partnership. Over the last two decades, legislative [42], executive [43], and payor-led [44] efforts have resulted in widespread adoption of electronic health records (EHR) [45] and collection of personal health records (PHR) [46]. Additionally, implementation of the Patient Protection and Affordable Care Act (Public Law 111–148) has encouraged the collection of patient-centered outcomes in clinical practice [47], which is consistent with the experience of other developed countries that have implemented routine use of outcome measurements in mental health care [48] and found that patient-reported outcomes were valued as useful by both patients and their providers [49,50]. 

The next step to create this empowered patient-physician partnership is to enhance the capacity of primary care clinics and increase the knowledge base of providers. Due to constraints of time and financial resources, it is also essential that the need for additional personnel is minimized, the administrative structure is kept simple, and overall cost savings are attained. Hence, the use of patient-centered self-report assessments delivered in an electronic format via health-IT can help minimize cost, reduce provider burden, provide education and self-efficacy in patients through better recognition of symptoms and change, and enhance patient and provider engagement in treatment. Further, the pace of introducing these changes may need to be gradual, allowing for a bottom-up building process that encourages knowledge acquisition and long-lasting cultural change in busy clinical practices. In Table 1, we have described our alternative approach to what needs to be done and how to do it, along with some of the commonly faced challenges and potential solutions. 

Table 1 describes the components involved in the implementation of the PCP-First approach along with commonly faced challenges and potential solutions.

## 7. Operationalizing Our Approach: Making Screening for Depression the Sixth Vital Sign

To test the feasibility and overall effect of PCP-First in “real-world” primary care settings, we developed a comprehensive program aimed to increase the number of depressed patients who are diagnosed, increase the accuracy of those diagnoses, and improve clinical outcomes over time by using empirically supported interventions. To emphasize the importance of depression assessment and management, we conceptualized depression as the sixth vital sign and named our project VitalSign^6^. The aims of VitalSign^6^ are as follows:Routinely and consistently identify patients suffering from depression using self-report assessments.Accurately diagnose and adequately manage patients who screen positive on self-report assessments.Improve long-term outcomes for depressed patients as compared to historical controls.

In order to achieve these aims, the PCP-First model is comprised of the following components:ScreeningDiagnosisTreatment SelectionTreatment ImplementationTreatment Revision

Each of these components is achieved through the implementation of multiple elements that are geared towards removing the barriers to care and improving the adoption of evidence-based practices (see Figure 1). These elements are implemented in a phased manner, with the intent of maximizing the available resources and increasing the likelihood of successful implementation in the community setting. Analogous to the management of other chronic diseases, we have established a stepwise process to achieve the goals, or outcome metrics, associated with these elements, while engaging the patient in learning how to manage their disease and engaging the provider to deliver high-quality care. For example, we assessed the impact of the identification of patients suffering from depression by establishing the rates of screening as well as the rates of positive screens. We measured the rates of accurate diagnosis and adequate management by using a diagnostic checklist, as well as follow-up selection by the providers. To assess treatment outcomes, we focus not only on improvements in symptom severity, treatment adherence, and side effects (which are the core components of Measurement-Based Care), but also changes in functioning as well as likelihood of treatment continuation.

Figure 1 describes the elements and goals associated with the successful adoption of the Primary Care Provider, PCP-First approach in community settings.

## 8. Methods

To improve the efficiency of VitalSign^6^, we have taken advantage of advances in health information technology (health IT) by creating a tool in which, assessments of symptoms, side-effects, and adherence are recorded in an easy-to-search format, that providers may use alongside clinical information recorded in the EHR. Additionally, to integrate information from published literature with clinical information, along with the models proposed by other chronic illnesses [51], we developed a comprehensive training program for clinical providers and support staff, that includes training manuals, reading materials, online training modules, and in-person training sessions. We also offer curbside consultation and periodic rounding with providers. To individualize the training program and implementation process, we also developed an extensive set of pre-implementation assessments in order to ensure the program’s successful integration into the clinic’s existing workflow. 

### 8.1. VitalSign^6^ Software

VitalSign^6^ utilizes a point-of-care, web-based software program, VS6, to screen all patients for depression, using the Patient Health Questionnaire (PHQ)-2, which is available in both English and Spanish for patients aged 12 and older. For patients who screen positive on the PHQ-2, the VS6 program probes further for additional depressive symptoms using the 9-item version of the PHQ (PHQ-9) and associated factors (e.g., anxiety, substance use, productivity, etc.). The application includes a ‘follow-up plan’ page that clinicians use to review results and document diagnoses and treatment plans. During subsequent follow-up visits, the VS6 software program assists the primary care provider in monitoring symptoms over time and guides treatment planning and medical decision making using principles of MBC. For patients who screen negative, the software triggers the need for a re-screen in one year. The VS6 application is constantly undergoing upgrades and improvements based on clinic partners’ feedback and the desire to improve efficiency, accuracy, and quality of care.

### 8.2. Patient Health Record

The assessments from the VS^6^ software are electronically stored in a protected health information (PHI) secured fashion in the proprietary database, that is housed within a secured institutional firewall (similar to the security for EHR) and includes limited demographic information, as well as MBC assessments. Additionally, users have to set-up an individual profile to access patient health record with security requirements that are similar to the EHR access. Finally, an audit trail is created that allows monitoring of individuals who have accessed a given patient’s health record ensuring the confidentiality of these records.

### 8.3. Electronic Health Record and Interoperability

The participating clinics, that have electronic health record systems, have agreed to contribute the data stored in the EHR to the VS^6^ database. However, the long-term goal of this model would provide collaborating clinics’ EHRs with the ability to interface with the VS^6^ software in real-time. As we move towards this goal, the VS^6^ application has recently been optimized for integration and interoperability using SMART on FHIR resources. Specifically, SMART (Substitutable Medical Apps and Reusable Technology) allows for the development of medical apps that can run on any EHR system. FHIR (Fast Healthcare Interoperability Resources) uses a modern web-based suite of API technology for user interface integration, allowing it to facilitate interoperation between disparate health care systems. Blending the two together, SMART on FHIR allows VS^6^ to become integrated directly into the EHR user interface. Now, when using VS^6^, clinic staff can search for patient data directly in the EHR, eliminating the need for entering patient data in both systems, which saves time and improves data integrity. Providers are able to streamline their review of VS^6^ clinical data from their EHR. Subsequently, this allows them to reach a diagnosis and to develop a customized treatment plan, based on the clinical decision support algorithms for depression at the point of care. Using FHIR, which allows for the efficient and secure exchange of data between VS^6^ and the EHR, VS^6^ is also capable of retrieving patient diagnoses and medication data directly from the EHR, which allows providers to immediately see any changes that originate in the EHR, which could impact treatment decisions in VS^6^. Figure 2 depicts the SMART on FHIR integration workflow. These technological improvements to the application have the potential to improve depression treatment and overall patient outcomes.

### 8.4. Remote Patient Assessment

The VS^6^ application has recently added a new feature, Remote Patient Assessment (RPA), which allows the patient to complete the assessment instruments for measurement-based care from the privacy of their home via their own electronic devices (e.g., computer, cell phone) at prescribed intervals prior to the scheduled office visit. This feature can be enabled/disabled at the clinic and patient level. In a clinic that has elected to offer RPA, an invitation to participate is presented to the patient as they complete their measures in VS^6^. When the patient accepts, they enter their cell phone number and email address. VS^6^ immediately sends the patient a “welcome” email with a link to setup their user profile and password. At the next prescribed assessment date, the patient will receive notification via email or text message (patients’ selected preference) with a link to the patient portal, at which point they may log in and complete the measures. The measures are sent to the patient record in VS^6^ and an e-mail notification is sent to the clinic to let them know the measures have been completed, along with a link to the outcome screen for the patient’s measures. Clinical providers can then log in to view the patient outcomes at their convenience, which allows them to prepare for the patient’s visit ahead of time and improves clinic workflow efficiency. 

### 8.5. Selection of Collaborating Clinics

The selection of collaborating clinics, during the first phase of the project, was intended to maximize the representation of minority and under-served communities. To this end, primary care clinics, that predominantly serve uninsured or low-income minority populations, were approached to participate in the project. As part of initial conversations with potential clinic partners, the team clearly conveys that VitalSign^6^ is geared towards making the management of depression similar to other chronic illnesses such as hypertension and diabetes, where the bulk of the treatment is provided by the primary care providers, and engagement of specialists, like psychiatrists, is limited to difficult-to-treat or complicated cases. Further discussions are focused around the logistics of implementation, timeline, expectations from the collaborating clinic, and training and support provided by the VitalSign^6^ team, thereby placing emphasis on how routine measurements are critical to provide high-quality evidence-based treatment for patients. As part of site recruitment activities, the team makes a formal presentation to the clinical providers to highlight the existing gap in care and importance of routine measurements. This presentation is aimed to generate “buy-in” from the clinical providers, which in turn will contribute to the successful implementation of the project. 

### 8.6. Implementation of VitalSign^6^

Implementation is conducted in a phased manner, which includes the following steps. See Figure 3 for more details.
Workflow integration assessment, which allows the project team to customize implementation, based on clinical operations information (number of providers and exam rooms, flow of patients through the clinic, as well as existing practices for management of chronic illnesses).Depression education and application training for all clinic staff, via in-person and online modules, including an overview of depression and introduction to MBC.Launch planning, including finalization of workflow and set-up of individual profiles in the VS^6^ software application.Initiation of screening and MBC implementation with a “Soft Launch”. During the initial 2-week period, a member of the VitalSign^6^ team is onsite at the clinic, reviewing training material with clinic staff, facilitating incorporation of depression screening into the clinic’s workflow, and ensuring that all clinic staff are comfortable using the VS^6^ application and administering the screening. For those patients who screen positive, clinicians are instructed on how to conduct a clinical interview and document the follow-up plan. During this period and beyond, a consulting psychologist/psychiatrist on the VitalSign^6^ team is available to coach the clinicians on how to manage their depressed patients.

Figure 3 illustrates the phased approach to the implementation of the VitalSign^6^ project in a primary care setting.

### 8.7. Operationalization of VitalSign^6^ Components

The VitalSign^6^ project operationalizes the five core components of the PCP-First model, listed above, in real-world settings as follows:Screening—The initial screening for all patients is conducted through the VS^6^ software, with the two-item Patient Health Questionnaire (PHQ-2), which assesses the two core symptoms of major depression (sad mood and anhedonia) with items scored from 0–3 (PHQ-2 range: 0–6) [44]. A positive screen on the PHQ-2 is defined as a total score greater than 2 [44]. For those who screen positive on the PHQ-2, VS^6^ immediately generates the 9-item Patient Health Questionnaire (PHQ-9), which assesses for symptoms in all nine domains of a major depressive episode [45]. The standard package of additional assessments that patients receive with a positive PHQ-2 screen include (1) 7-item Generalized Anxiety Disorder (GAD-7) scale, which assesses for symptoms in seven domains of generalized anxiety disorder [52]; (2) 5-item Altman Self-Rating Mania Scale (ASRM), which assesses for symptoms of mania [53]; (3) 2-item Physical Activity Questionnaire, which assesses patient’s level of physical activity [54]; (4) 4-item Pain Frequency, Intensity, and Burden Scale (P-FIBS), a brief, self-administered measurement of pain frequency, intensity, and burden [55]; (5) single-item screening question each for alcohol and drug use disorders [56,57], each designed to screen for current usage patterns that might impact depression treatment; (6) 6-item Work Productivity and Activity Impairment Questionnaire (WPAI), which assesses to what extent the patient’s depressive symptoms impacted his/her work attendance and productivity [58]; and (7) the 16-item Concise Associated Symptoms Tracking—Self-Report Scale (CAST-SR), which assesses for the five known treatment-emergent symptom domains: Irritability, anxiety, mania, insomnia, and panic [59,60]. For those with a negative PHQ-2 screen, no further assessment is mandated. As soon as a patient completes her/his assessment, the results are immediately available for review by the clinician.Diagnosis—Clinicians have the option to administer optional self-report assessments, available within VS^6^, to collect additional information needed to conduct a thorough diagnostic assessment. For example, in patients with positive PHQ-2 screen, their response to the 9th item of PHQ-9 (“Thoughts that you would be better off dead or of hurting yourself in some way”) is flagged for clinicians. Clinicians have the option to administer additional validated instruments for detailed suicide risk assessments, including the Concise Health Risk Tracking (CHRT) scale [61], which has been validated in diverse groups of patients with psychiatric disorders [62,63,64]. Additionally, clinicians are trained on identifying the risk factors for suicide and on utilizing appropriate triage pathways based on the resources available. Such an approach is consistent with recommendations for depression screening in other medical settings, such as in patients with cardiovascular disease [65]. Also, for patients who screened positive for alcohol or drug use, providers may wish to administer the 28-item Drug Abuse Screening Test (DAST) [66] or the 24-item Michigan Alcoholism Screening Test (MAST) [67]. Clinical providers can then utilize the Major Depression Diagnostic Checklist based on the Diagnostic and Statistical Manual 5th edition (DSM-5), available within VS^6^ on the follow-up plan page, to confirm or rule out a diagnosis [68]. Clinicians also have the option to conduct additional follow-up visit(s) to ensure accurate diagnosis. Treatment Selection—The treatment plan options available to providers include: (1) pharmacological treatment implemented with MBC; (2) active surveillance by the primary care clinician with symptomatic monitoring; (3) behavioral treatment, such as psychotherapy, in primary care setting, through integrated behavioral health providers; (4) exercise; (5) referral to an external provider for specialty care; or (6) no further follow-up. Additionally, clinicians could indicate whether the patient refused the prescribed treatment option. Primary care clinicians are predominantly trained on pharmacotherapy for depression. However, training, as well as referral resources for other evidence-based treatment for depression, such as psychotherapy and exercise, are readily available.Treatment Implementation—The adherence of depressed patients to prescribed antidepressant treatment is measured using the Patient Adherence Questionnaire (PAQ) [69], which is a two-item self-report instrument administered via VS^6^. Patients who do not take their prescribed medications more than 70% of the time are considered non-adherent. For these patients, the instrument asks them to select the reasons for their non-adherence and these reasons are monitored by the prescribing clinicians. Prescribing clinicians also monitor medication side-effects, using the Frequency, Intensity, and Burden of Side Effects Rating (FIBSER) scale [70], which is a three-item, self-report instrument that assesses the frequency, intensity, and functional impairment, or burden, of reported side effects. The VS^6^ application scores the measures and reports the results to the provider. Based on the score, the side effects are categorized as acceptable, requiring attention, or unacceptable.Treatment Revision—VitalSign^6^ provides electronic clinical decision support for Measurement-Based Care treatment of depression at the point of care. It facilitates the delivery of personalized care, by using data from self-report measures, completed by the patient during the visit before being seen by the provider. VitalSign^6^ sends all of the relevant data points to its sophisticated logics engine which contains all the intelligence of the best practice treatment algorithms for treatment of depression. The logics engine uses the patient’s individual measures data to calculate the most appropriate treatment recommendation specific to the current point of care.

## 9. Metrics of Success

The RE-AIM framework is used to assess the impact of VitalSign^6^ across the following five factors: Reach (what proportion of the target population participated?), Efficacy (what is the impact on specified outcome criteria?), Adoption (what proportion of practices/clinicians will adopt this program?), Implementation (what is the quality/consistency of delivery in real-world settings?) and Maintenance (to what extent is the program sustained over time?). The metrics associated with each factor, which are calculated for the project as a whole, as well as for each clinic, are described below. See Table 2.

**Reach**. The reach of VitalSign^6^ is measured in terms of participation of staff and clinicians at the participating clinics along with the patients seen at those clinics. The number of staff and clinicians participating in the project annually constitutes the numerator, and the total number of staff and clinicians serve as the denominator at each clinic. To evaluate the reach at the patient-level, a screening rate is calculated with the numerator as the number of patients screened and the denominator as the number of total patients seen at the clinic. The screening rate is calculated on an annual basis. A screening rate ≥ 75% is considered the metric of success.

**Efficacy**. The primary outcome of interest is attainment of symptomatic remission in those diagnosed with a depressive disorder (MDD, Persistent Depressive Disorder, Adjustment Disorder with Depressed Mood, Unspecified (i.e., subclinical) Depressive Disorder and initiated in MBC. The rates of remission are calculated for the acute-phase treatment (18 weeks post treatment-initiation) as well as for the long-term (after 1 year). Remission rates ≥ 25% are considered as the metric of success. The efficacy measures also include effects of depression screening and treatment on comorbid medical conditions by evaluating outcomes, collected as part of routine clinical care. While no formal metrics have been decided a priori for these outcomes, due to heterogeneity of data collected as part of routine care, exploratory analyses will be conducted to evaluate whether remission of depression symptoms improves outcomes of medical comorbidities or vice versa.

**Adoption**. As VitalSign^6^ implementation is customized to each clinic for the smooth integration into the clinic workflow, the members of the study team will conduct semi-structured interviews with staff and clinicians at each clinic to evaluate the adoption of depression screening and MBC implementation, in those who screen positive. These interviews will also identify barriers to adoption and steps taken to overcome those barriers. An initial evaluation of VitalSign^6^ via semi-structured interviews with clinicians was conducted, and results indicated that despite barriers, especially those related to time, the program was successful in improving providers’ knowledge, attitudes, and skills related depression screening and treatment [71].

**Implementation**. We evaluate the implementation of the project by calculating how often depressed patients who were initiated into MBC completed the assessments in VS^6^ upon return to the clinic. We will also evaluate the sociodemographic and clinical characteristics of patients who do not return for visits and follow-up assessments, once initiated into MBC, through individual interviews with patients.

**Maintenance**. The extent to which remission persists over a prolonged period of time (5 or 10 years) will serve as a marker for maintenance of the effects at the individual patient level. At the program level, follow-up surveys and semi-structured interviews will be conducted to establish the extent to which universal screening for depression and evidence-based treatment in those who are diagnosed with depression is sustainable and becomes a part of the clinic’s culture over time.

The Reach, Efficacy, Adoption, Implementation, and Maintenance (RE-AIM) framework was used to evaluate the impact of the implementation of Primary Care First model in the VitalSign^6^ project. PHQ-9 is Patient Health Questionnaire 9-items. MBC is Measurement Based Care, VS6 refers to the software used in the VitalSign^6^ project.

## 10. Discussion

The VitalSign^6^ project addresses the unmet need of under-identification and under-treatment of depressive disorders, especially in resource-poor clinics that care for underserved and minority populations. By conceptualizing depressive disorders as chronic medical conditions and utilizing health IT advances, this project has the potential to transform “treatment as usual” into evidence-based, high-quality care in primary care settings. By design, the project’s PCP-First approach does not add personnel resources beyond those already serving in a clinic. This bare-bones approach can be augmented by other models of care. For example, clinics participating in the VitalSign^6^ project have already implemented mental health navigation [72,73] and behavioral activation teletherapy [74,75]. This approach also offers the potential to add-on research and quality improvement projects. The accuracy of clinician diagnostic assessments can be ascertained by conducting structured diagnostic interviews, with independent raters in a subset of the sample, and comparing those to the diagnoses made by treating clinical providers. The patient self-report assessments, collected as part of the VitalSign^6^ project, can also be used as part of longitudinal observational studies [such as UTSW Depression Cohort: A Longitudinal Study of Depression (NCT NCT02697487, Dallas 2K: A Natural History Study of Depression (D2K, NCT02919280), and Resilience in Adolescent Development (RAD, NCT03458936)], along with research-only assessments and collection of biospecimens. The quality improvement projects include, interventions conducted by VitalSign^6^ team members as part of UT Southwestern’s Clinical Safety and Effectiveness course aimed at improving selected participating clinics’ use of MBC for patients with depression. Additional quality improvement projects include benchmarking of clinics’ performance against each other. 

There are several limitations to the findings from the VitalSign^6^ project. The generalizability may be limited, as clinics selected as part of this project predominantly serve minority and un- or under-insured populations. Furthermore, as self-report measures in the VitalSign^6^ project are available only in English and Spanish, the findings of this project are limited to predominantly English and Spanish speaking populations. Additionally, as data was collected from real-world clinical populations, the number of assessments were kept to a minimum, thus, limiting detailed diagnostic and functional assessments. Also, as the screening efforts were focused on identifying patients with depression, other mental health problems may be missed out. For example, patients with current suicidality who do not have significant sad or depressed mood or anhedonia may be missed, as we currently do not offer additional screening assessments in those who screen negative on PHQ-2. Thus, based on clinical needs, additional screening for suicidality may be added in future [76]. Finally, the frequency of self-report assessments is limited by patients’ attendance at primary care clinic visits. A greater uptake of Remote Patient Assessment, via smartphone apps and web-based surveys, can increase the frequency of data collection and provide greater information about the course of treatment between clinic visits. 

Despite these limitations, the PCP-First model has the potential to reduce the barriers faced by healthcare systems attempting to improve routine depression screening and follow-up care in primary care settings. While the initial outcomes of the VitalSign^6^ project are discussed in a separate report [77], there is still much to be learned about how VitalSign^6^ can continue to evolve, through technology innovations, augmentation with other models of care, and implementation among diverse clinic models and patient populations, in order to maximize feasibility, sustainability, and patient outcomes. 

## Figures and Tables

**Figure 1 pharmaceuticals-12-00071-f001:**
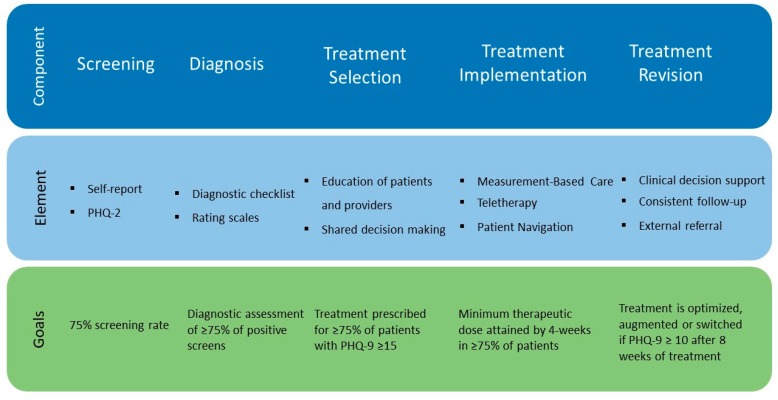
PCP-First: A phased approach to adoption and achievement.

**Figure 2 pharmaceuticals-12-00071-f002:**
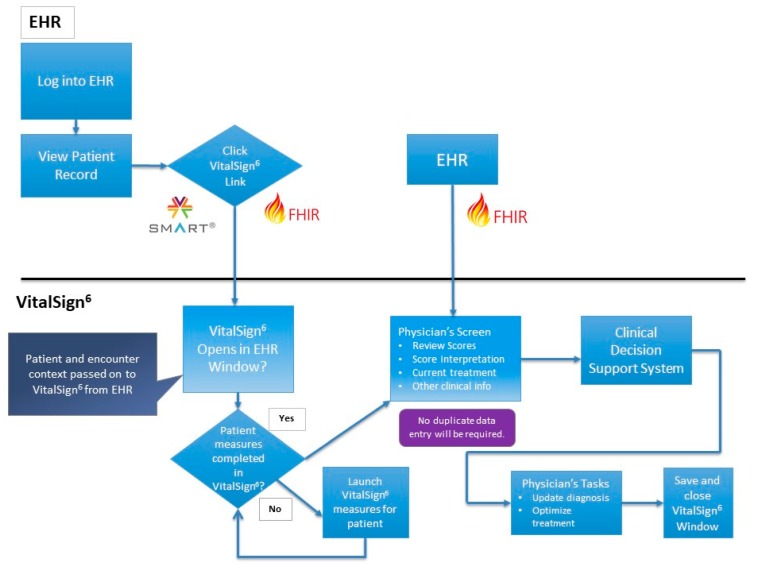
illustrates the practical application of the SMART (Substitutable Medical Apps and Reusable Technology) on FHIR (Fast Healthcare Interoperability Resources) technology, which allows VS^6^ to become integrated directly into the electronic health record (HER) user interface.

**Figure 3 pharmaceuticals-12-00071-f003:**
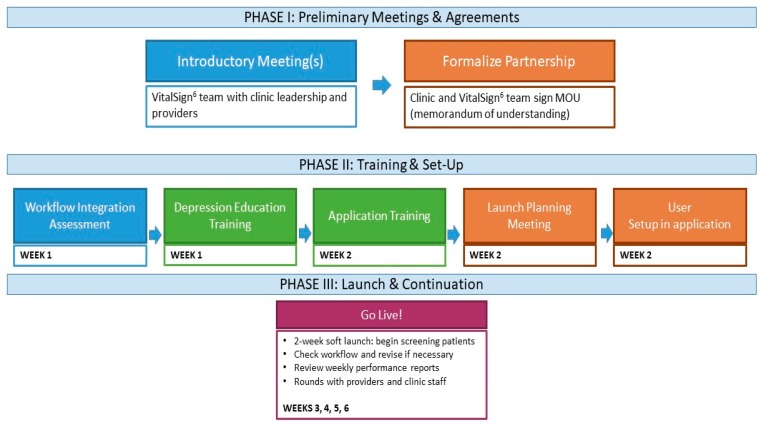
VitalSign^6^ Launch Process.

**Table 1 pharmaceuticals-12-00071-t001:** The Primary Care First (PCP)-First Approach to the Treatment of Depression.

Component	Clinical Tasks	Methods	Challenges	Potential Solutions
**Screening**	Detect depression	Administer PHQ-2	Documented on paper	Document directly in electronic health record (EHR)
			Results not readily available to providers	Results routed directly to providers in EHR
			Positive PHQ-2 should be followed by PHQ-9	Screen automatically expands to PHQ-9
			Repeat screens as depression is episodic	Negative screens are re-screened annually
**Diagnosis**	Confirm or rule out depressive disorder	DSM-5 criteria-driven diagnostic interview	Lack of comfort with diagnostic interview	Online and in-person training
			Diagnose based on overall clinical impression	Use DSM-5 checklist embedded in EHR
			Specialist input needed for complicated cases	Access to consulting clinicians and referral sources
**Treatment Selection**	Shared decision-making options: □ Active surveillance □ Medication □ Psychotherapy □ Exercise Combination	Provider training and patient education	Frequent in-person visits for active surveillance	Remote assessments and provider review in EHR
			Lack of comfort with prescribing antidepressants	Online and in-person training
			Limited access to evidence-based psychotherapy	Tele-health programs for psychotherapy
			Limited knowledge of exercise prescription	Consultation with exercise specialists
			Optimize pharmacotherapy and psychotherapy	PCPs closely collaborate with tele-health therapist
**Treatment Implementation**	Deliver treatment Measure outcomes Assess response	Measurement-Based Care (MBC)	Assess improvement with treatment	Validated measures of symptom and functioning
			Limited time for clinician assessments	Use of self-report assessments
			Poor adherence to prescribed treatment	Systematically assess adherence at each visit
			Side-effects results in treatment discontinuation	Systematic assessment of side effects at each visit
			Inability to find previous paper forms	Easily searchable results in an electronic format
			Unable to visualize changes over time	Custom reports for outcomes over time
			Patient barriers prevent consistent follow-up	Implement patient navigation programs
**Treatment Revision**	Based on response	Clinical Decision Support System	How to handle treatment-resistant depression?	In-person or phone consultation; refer to specialist

**Table 2 pharmaceuticals-12-00071-t002:** The operationalization of RE-AIM framework to measure success of the VitalSign^6^ project.

	Indicator	Metric
**Reach**	Clinician/staff participation	Number participating/total number of clinicians/staff at clinic
	Patient participation (screening rate)	Number screened/total number of unique patients at clinic
**Efficacy**	Remission rates	PHQ-9 score < 5: acute-phase (18 weeks); long-term (1 year)
	Impact on comorbid medical conditions	Exploratory analyses
**Adoption**	Adoption of depression screening and MBC implementation	Semi-structured clinician and staff interviews
**Implementation**	Completion of MBC measures at follow-up visits	Number who completed follow-up measures/total number of patients who were due for follow-up assessments
	Characteristics of patients who do not return for visits (“lost to care”)	Out-reach and semi-structured interviews
**Maintenance**	Patient-level sustainability	Sustained remission over 5 or 10 years
	Program-level sustainability	Follow-up surveys and semi-structured interviews

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
