# Peer review of "VitalSign6: A Primary Care First (PCP-First) Model for Universal Screening and Measurement-Based Care for Depression"

_pharmaceuticals, 2019, doi:10.3390/ph12020071_

Round 1
Reviewer 1 Report
The work from Trivedi et al, entitled “VitalSign6: A Primary Care First Model for Universal 2 Screening and Measurement-Based Care for 3 Depression” This is an interesting article of unique relevance in the field, on quality improvement measures that aims on incorporating information technology through integrated primary care model. Their aim is to provide medical providers with proper means to efficiently and effectively diagnose and treat depression in primary care and other specialty settings. In an eloquent and systematic manner authors provide steps that would improving vital components of screening, detection, proper diagnosis and appropriate treatment in depression.
Minor points:
Recognizing the magnitude of the current suicide crisis in the US and depression as its main underlying cause I wonder if screening for suicide on its own rather that just using items of PSQ-9 would be possible. For example a recent effort spearheaded by NIMH and validated in various clinical setting show that using the Ask Suicide-Screening Questions (ASQ), which consists of four yes/no questions and takes only 20 seconds to administer improves detection of suicide in various clinical settings (see Horowitz et al., 2013 Psychosomatics).
Can authors comment on this and describes if Vitalsign6 has sensitivity or specificity in suicide detection?
Author Response
REVIEWER 1:
Comments and Suggestions for Authors:
The work from Trivedi et al, entitled “VitalSign6: A Primary Care First Model for Universal 2 Screening and Measurement-Based Care for 3 Depression” This is an interesting article of unique relevance in the field, on quality improvement measures that aims on incorporating information technology through integrated primary care model. Their aim is to provide medical providers with proper means to efficiently and effectively diagnose and treat depression in primary care and other specialty settings. In an eloquent and systematic manner authors provide steps that would improving vital components of screening, detection, proper diagnosis and appropriate treatment in depression.
Authors’ response: Thanks for your positive feedback on our manuscript.
Minor points:
Recognizing the magnitude of the current suicide crisis in the US and depression as its main underlying cause I wonder if screening for suicide on its own rather that just using items of PSQ-9 would be possible. For example a recent effort spearheaded by NIMH and validated in various clinical setting show that using the Ask Suicide-Screening Questions (ASQ), which consists of four yes/no questions and takes only 20 seconds to administer improves detection of suicide in various clinical settings (see Horowitz et al., 2013 Psychosomatics).
Can authors comment on this and describes if Vitalsign6 has sensitivity or specificity in suicide detection?
Authors’ response: Thanks for bringing up the issue regarding suicide detection. We have clarified that in patients who screen positive, response to the 9th item of PHQ-9 which asks about suicidality is flagged for the clinician to review (Page 11 Para 2). Additionally, we have expanded on the use of Concise Health Risk Tracking (CHRT) scale which is included as an optional self-report measure in the VitalSign6 program. During their training and over subsequent consultations, clinicians are trained on suicide risk assessment so that they can safely manage suicidality in their patients. Finally, we have acknowledged as a limitation that patients with suicidality who screen negative on PHQ-2 may be missed (Page 14 Para 2). Future considerations for the VitalSign6 project may include addition of suicidality screen for patients who screen negative on PHQ-2 based on the needs of individual clinics and the burden of additional assessments on patients.
Reviewer 2 Report
Dear authors, your manuscript addresses a very relevant topic such as the screening and treatment of depression within a primary care setting. I enjoyed reading this work, but I think it would be relevant to discuss also how the safety of the patient (e.g., suicidal thoughts) would be monitored, especially in case of Remote Patient Assessment, and which are the implications of this extensive web application in terms of patients' confidentiality.
Author Response
REVIEWER 2:
Comments and Suggestions for Authors:
Dear authors, your manuscript addresses a very relevant topic such as the screening and treatment of depression within a primary care setting. I enjoyed reading this work, but I think it would be relevant to discuss also how the safety of the patient (e.g., suicidal thoughts) would be monitored, especially in case of Remote Patient Assessment, and which are the implications of this extensive web application in terms of patients' confidentiality.
Authors’ response: Thanks for bringing up the concerns regarding safety assessments. As we mentioned in our response to Reviewer 1’s comment, the 9th item of PHQ-9 is flagged for the clinician’s review (Page 11 Para 2) and optional assessments can be conducted. Additionally, clinicians are trained on suicide risk assessments and educated about the triage pathways available. Regarding confidentiality, we have clarified (Page 8 Para 3) security of web application is similar to that of EHR and that audit trails are created to ensure that only clinicians involved in care of a patient are accessing that information.